# LEARNING ACTIVATION FUNCTIONS WITH PCA ON A SET OF DIVERSE PIECEWISE-LINEAR SELF-TRAINED MAPPINGS

## ABSTRACT

This work explores a novel approach to learning activation functions, moving beyond the current reliance on human-engineered designs like the ReLU. Activation functions are crucial for the performance of deep neural networks, yet selecting an optimal one remains challenging. While recent efforts have focused on automatically searching for these functions using a parametric approach, our research does not assume any predefined functional form and lets the activation function be approximated by a subnetwork within a larger network, following the Network in Network (NIN) paradigm. We propose to train several networks on a range of problems to generate a diverse set of effective activation functions, and subsequently apply Principal Component Analysis (PCA) to this collection of functions to uncover their underlying structure. Our experiments show that only a few principal components are enough to explain most of the variance in the learned functions, and that these components have in general a simple, identifiable analytical form. Experiments using the analytical function form achieve state of the art performance, highlighting the potential of this data-driven approach to activation function design.

## 1 INTRODUCTION

Deep learning, a powerful branch of machine learning, is the backbone of modern artificial intelligence. It has achieved remarkable success in various domains, from computer vision to drug discovery and autonomous vehicles, by learning intricate patterns and representations from massive datasets. This capability stems from the architecture of deep neural networks (DNNs), which are built with multiple layers of interconnected nodes. At the root of each of these nodes lies a crucial component: the activation function.

Activation functions introduce non-linearity into a neural network, which is essential for solving complex, real-world problems. Without them, a neural network would simply be a linear model. To provide a foundational basis for our work, we first review the essential properties and historical context of these functions. The two most critical requirements are non-linearity and differentiability. The primary and most indispensable purpose of an activation function, denoted as $\sigma$, is to introduce non-linearity. This fundamental requirement is formally justified by the Universal Approximation Theorem. Pioneering work by Cybenko (Cybenko, 1989) and Hornik (Hornik et al., 1989; Hornik, 1991) established that a feed-forward network with a single hidden layer can approximate any continuous function to a desired degree of precision, provided the activation function is non-linear. The mere existence of a non-linear function is a sufficient condition to allow for powerful function approximation. Beyond non-linearity, differentiability is a crucial property that enables gradient-based optimization. The standard backpropagation algorithm (Rumelhart et al., 1986) updates network weights by computing the gradient of the loss function via the chain rule. The activation function must therefore have a well-defined derivative to act as a backwards-traversable link in the network's computational graph. While strict differentiability is ideal, piecewise differentiability is sufficient in practice. Furthermore, the smoothness of an activation function, related to its higher-order differentiability, can significantly impact training dynamics by producing a smoother loss landscape, which is generally easier for optimization algorithms to navigate (Santurkar et al., 2018; Misra, 2019).

The history of activation functions reflects the evolution of deep learning itself. Early networks like the Perceptron (Rosenblatt, 1958) used a simple step function, which was non-differentiable and thus incompatible with gradient-based learning. With the popularization of the backpropagation algorithm (Rumelhart et al., 1986; Linnainmaa, 1976; Werbos, 2005), smooth and differentiable functions like the sigmoid and tanh became the standard. However, their saturating nature led to the *vanishing gradient* problem, which severely hampered the training of very deep networks. The Rectified Linear Unit (ReLU) (Nair & Hinton, 2010) solved this problem for positive activations and was computationally inexpensive, becoming a key factor in the early successes of deep learning. Since then, numerous variants have been proposed to address ReLU's *dying neuron* problem, such as Leaky ReLU (LReLU) (Maas et al., 2013), which introduced a small non-zero slope to the negative part of the function. This trend led to Parametric ReLU (PReLU) (He et al., 2015), which generalized LReLU by making the slope a learnable parameter. More recently, functions like Swish (Ramachandran et al., 2017) and Mish (Misra, 2019) have demonstrated that non-monotonicity can enhance model expressivity and improve gradient flow. A crucial aspect of these functions, which were discovered using automated search techniques, is that they empirically outperform other functions, showing that there could be better options than those designed by humans. The computational cost of activation functions is also a critical practical concern, as simple functions relying on inexpensive hardware operations (e.g., ReLU) are significantly faster than those involving costly operations like exponentials (e.g., Sigmoid, Tanh) (Datta, 2020).

A notable trend in recent years has been to move away from manually designed functions toward learnable activation functions, whose shapes are optimized during training. This evolution mirrors a broader trend in machine learning, reflecting a shift from manual engineering to automated discovery. The central motivation of this work is to formulate and study the behavior of an adaptive activation function whose parameters are learned dynamically during a neural network's training process. While traditional, static functions like ReLU and sigmoid have driven significant progress, they operate under a fundamental limitation: their form is fixed before training begins. This static nature can hinder the network's ability to capture complex, non-linear relationships and may lead to issues like vanishing gradients or slow convergence. To address these shortcomings, several strategies have been proposed. One approach extends existing functions with learnable parameters, as seen in PReLU (He et al., 2015) and the original trainable version of Swish (Ramachandran et al., 2017). Another method uses a linear combination of basis functions to construct a more complex, adaptable shape, a technique used in Adaptive Piecewise Linear (APL) Units (Agostinelli et al., 2014) and kernel-based approaches (Scardapane et al., 2019). A third line of research has focused on redesigning the network's internal components to create implicitly learned activation mechanisms. This includes the Network in Network (NIN) architecture (Lin et al., 2013), where a small multi-layer perceptron (MLP) replaces the standard activation, and Maxout networks (Goodfellow et al., 2013), which learn a piecewise linear activation.

Motivated by this trend, we propose a new approach by exploring the concept of a Self-Learning Activation Function (SLAF). Unlike parametric search methods that assume a predefined functional form, we model the SLAF as a small neural network—specifically, a single-layer MLP. This approach, which closely resembles the one presented in NIN (Lin et al., 2013) and the APL units in (Agostinelli et al., 2014), allows the network to learn the optimal non-linearity in a less biased way, as the function's parameters are learned and adjusted dynamically alongside the network's weights and biases using backpropagation. However, the main contribution of this work is not the SLAF itself, but a novel, straightforward method for discovering new activation functions from several SLAFs trained on a variety of problems. We do not perform any parametric search like in similar approaches (Ramachandran et al., 2017). Instead, we train a diverse collection of networks to learn a wide variety of SLAF instances. We then perform a Principal Component Analysis (PCA) on these learned functions to identify their main functional modes. By fitting simple analytical functions to these principal components, we are able to derive a new, powerful activation function, which we term *twish*. This approach simplifies the search process and provides a more generalizable method for discovering functional activation functions. Our results demonstrate that the *twish* function, which is a generalization of the Swish, consistently outperforms other popular activations and leads to faster convergence, particularly on more complex datasets. Our findings demonstrate that a data-driven approach to function discovery provides a powerful foundation for developing more sophisticated and robust activation functions for neural networks.

## 2 LEARNING THE BASE ACTIVATION FUNCTIONS

In this section, we first define the form of our activation function as a small network within the main network, and then we detail the procedure that is followed to obtain a diverse collection of basic activation functions. This set of activations will be the basis for the subsequent principal component analysis.

### 2.1 SLAF DEFINITION

The core of this research is a small feedforward network that implements a self-learnable activation function (SLAF). Because activation functions are pointwise operations, the SLAF is designed to take a single scalar input and produce a single scalar output. The SLAF is structured as a one-hidden-layer network that maps its scalar input to a higher-dimensional latent space before collapsing it back into a single output value. More specifically, the SLAF's single scalar input is transformed by a dense linear layer that maps it to $N$ hidden units, using weights $W_1 \in \mathbb{R}^{1 \times N}$ and bias $b_1 \in \mathbb{R}^N$. A ReLU activation function is then applied, and an output dense layer maps the $N$ hidden units back to a single output value, with weights $W_2 \in \mathbb{R}^{N \times 1}$ and bias $b_2 \in \mathbb{R}$. Figure 1 provides a visual representation of the SLAF for $N = 2, 4, 8$.

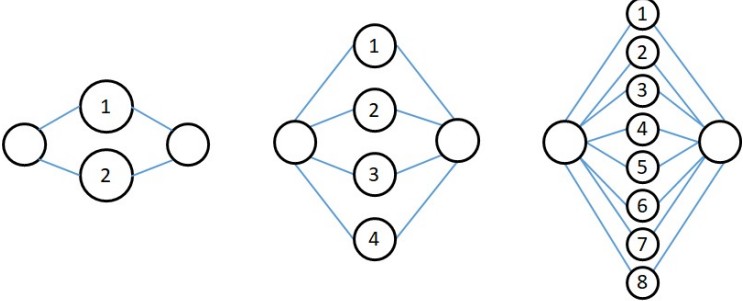

Figure 1: A diagram of SLAF-2, SLAF-4 and SLAF-8.

This definition can be understood as a piecewise linear function with as many pieces as hidden units. The intuition is that the final unit is a weighted sum of a finite number of ReLU-like functions. In the end the expression is

$$f(x) = \sum_i W_{2i}\text{ReLU}(W_{1i}x + b_{1i}) + b_2. \tag{1}$$

This is very similar to the piecewise linear function definition of the APL activation function in Agostinelli et al. (2014).

### 2.2 TRAINING

The parameters of the SLAF are trained jointly with those of the main network in which it is embedded. Our initial experiments use feed-forward neural networks trained on the MNIST, Fashion-MNIST, and CIFAR-10 datasets. The networks consist of two densely connected layers. They first reduce the input dimensions from $H \times W \times C$ to a 64-dimensional hidden state before finally generating a 10-dimensional vector of logits for classification. Given the varying input shapes ($28 \times 28 \times 1$ for MNIST and FashionMNIST, and $32 \times 32 \times 3$ for CIFAR-10), the networks have approximately 51K trainable parameters for MNIST-like images and 66K for CIFAR-10 images.

For each dataset and for each number of SLAF units, $N \in \{2, 4, 8, 16, 32, 64\}$, we trained 256 different networks. Each network was trained for a fixed number of 10 epochs using the Adam optimizer (Kingma & Ba, 2014) to minimize a cross-entropy loss between the output probabilities and the expected targets. To ensure comparable input scales, pre-activations were batch-normalized before applying the SLAF activations. A random sample of the learned activation functions for $N = 16$ is shown in Figure 2. The $y$-axis scale varies for each plot to facilitate easier comparison of

the function shapes. We observe that the learned shapes are similar, though diverse, across different problems. This is also true for different values of $N$ (not shown).

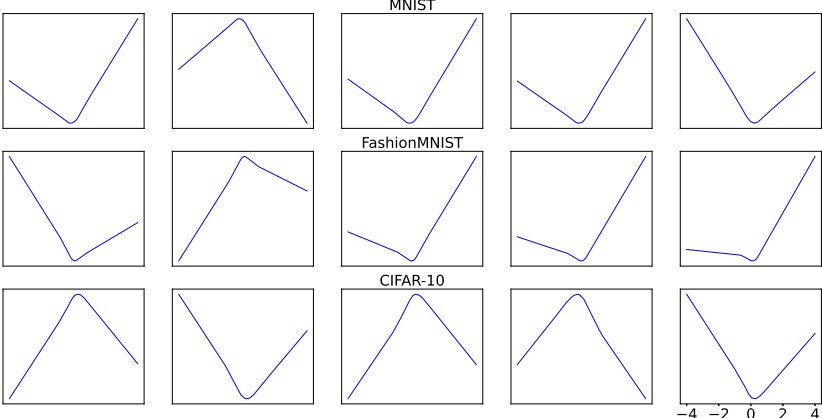

Figure 2: Sample SLAFs obtained for $N = 16$ for the datasets MNIST (top), FashionMNIST (middle) and CIFAR-10 (bottom).

## 3 ACTIVATION FUNCTION DISCOVERY THROUGH PCA

The main idea of this work is to use the set of trained activation functions as the basis for a Principal Component Analysis (PCA) that allows to identify the main functional modes. Therefore, our next step is to perform PCA to identify the functional shapes that account for the most variance in the SLAF functions. For this analysis, we used all $4608$ trained activation functions (3 datasets $\times$ 6 SLAF functions $\times$ 256 networks). To perform the PCA, each SLAF was evaluated at $1600$ equally spaced points in the range $[-4, 4]$. The analysis yielded a series of eigenfunctions that represent the primary modes of variation in the function shapes. The first two principal components, which jointly explain more than $99.5\%$ of the total variance, are shown in Figure 3. Interestingly, these two components can be approximated by simple analytical functions.

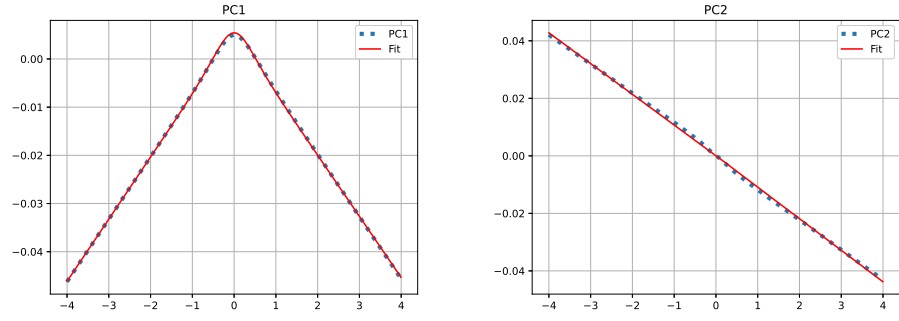

Figure 3: First two eigenfunctions obtained with PCA on the trained SLAFs.

The first principal component (PC1) is symmetric with respect to the $y$-axis, and consists of two linear branches with a smooth transition around $x = 0$. Up to a multiplicative constant, this shape can be understood as a soft absolute value function. Ignoring this scale factor and an additive bias, it can be formally approximated by

$$f_1(x) = x \tanh(\beta x). \tag{2}$$

The original eigenfunction (blue, dotted) as well the approximation (red, solid), are shown in figure 3 (left panel). The second principal component (PC2), on the other hand, is essentially a linear function of $x$ with no bias:

$$f_2(x) = \gamma x. \tag{3}$$

The right panel of figure 3 shows both the experimental eigenfuntion (blue, dotted) and its analytical approximation (red, solid).

### 3.1 THE TWISH ACTIVATION FUNCTION

As previously noted, more than $99.5\%$ of the variance in the shape of the SLAFs is explained with just the first two eigenfunctions. Even more, these two principal components can be well approximated by the expressions in equations 2 and 3. This suggests that the general functional shape of the learned activations can be characterized as a parametric function of the form:

$$f(x; \beta, \gamma) = x \tanh(\beta x) + \gamma x, \tag{4}$$

where $\beta$ and $\gamma$ are learnable parameters. It is interesting to note that, for $\gamma = 1$, the function resembles the Swish activation function (Ramachandran et al., 2017):

$$f(x; \beta, \gamma = 1) = 2x\sigma(2\beta x). \tag{5}$$

However, our learned function, which we term *twish*, generalizes this form. Unlike Swish, the twish function incorporates a wider variety of shapes for $\gamma \neq 1$ (see figure 4), effectively having the Swish as a particular case. We named it *twish* because the functional form of its first principal component is similar to the Swish function, but uses a hyperbolic tangent instead of a sigmoid. In the following section, we describe additional experiments where we apply the twish function to the previous datasets using convolutional neural networks, comparing its performance against other state-of-the art activations.

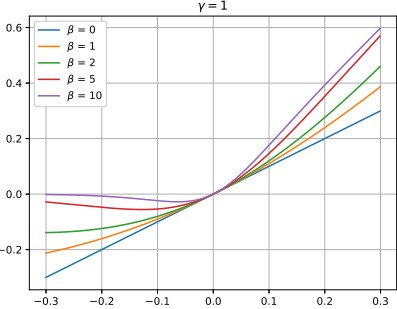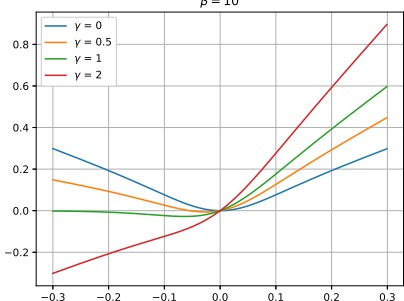

Figure 4: The twish function.

## 4 FURTHER EXPERIMENTS

To evaluate the potential benefits of the newly described twish activation function, we compare it with other commonly used functions: ReLU, pReLU, Swish, ELU, SELU, GELU and Mish. For this comparison, we use the same three datasets, namely MNIST, FashionMNIST, and CIFAR-10, with a custom convolutional neural network (CNN). In the case of the Swish and twish functions, the trainable parameters are shared across all the network's neurons. The Swish $\beta$ parameter is

Table 1: Test accuracy at the end of the 30 training epochs.

| AF | MNIST | FashionMNIST | CIFAR-10 |
|---|---|---|---|
| relu | $0.989 \pm 0.001$ | $0.905 \pm 0.003$ | $0.693 \pm 0.008$ |
| prelu | $0.989 \pm 0.002$ | $0.905 \pm 0.004$ | $0.702 \pm 0.007$ |
| elu | $0.987 \pm 0.001$ | $0.902 \pm 0.003$ | $0.679 \pm 0.006$ |
| selu | $0.986 \pm 0.001$ | $0.900 \pm 0.004$ | $0.662 \pm 0.006$ |
| gelu | $0.989 \pm 0.002$ | $0.904 \pm 0.003$ | $0.697 \pm 0.007$ |
| mish | $\mathbf{0.990 \pm 0.001}$ | $0.903 \pm 0.003$ | $0.693 \pm 0.006$ |
| swish | $\mathbf{0.990 \pm 0.001}$ | $0.905 \pm 0.003$ | $0.695 \pm 0.006$ |
| twish | $\mathbf{0.990 \pm 0.002}$ | $\mathbf{0.906 \pm 0.004}$ | $\mathbf{0.705 \pm 0.007}$ |

initialized to 1, and the twish parameters are initially set so that the starting point is also a Swish with $\beta = 1$.

The CNN has a simple architecture consisting of two convolutional layers with $3 \times 3$ kernels, each followed by a max-pooling layer to halve the image resolution. The number of channels increases from 1 (for grayscale images) or 3 (for RGB images) to 16, and then to 32. After the two convolutional and pooling layers, the output is flattened before the final linear transformation to a 10-dimensional vector for classification. The size of this final fully connected layer differs between the datasets (MNIST/FashionMNIST vs. CIFAR-10) due to their varying initial image dimensions. The networks have approximately 20K and 25K trainable parameters for MNIST-like and CIFAR-10 images, respectively. As before, batch normalization is applied to all pre-activations. For each dataset and activation function, we train 64 networks for 30 epochs using the Adam optimizer (Kingma & Ba, 2014), and then average the results.

Figure 5 shows the test set accuracy as a function of the training epoch for CNNs trained on MNIST (left panel) and CIFAR-10 (right panel). The curves represent the average accuracy over the 64 networks trained for each activation function. We observe that the twish function clearly dominates in both cases, outperforming the other activations by a wide margin throughout the entire training period. This superiority is consistent across both datasets, despite the overall difference in accuracy between the two problems. For a more detailed analysis, we refer the reader to Tables 1 and 2.

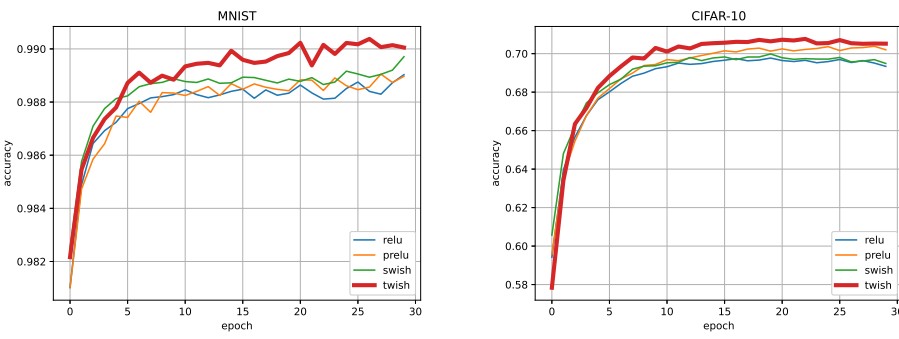

Figure 5: Test accuracy vs. training epoch for MNIST (left) and CIFAR-10 (right).

Table 1 displays the average test accuracy obtained after 30 training epochs for each of the three datasets. We observe that the twish function yields the best results in all cases. The performance difference is especially significant for the CIFAR-10 dataset, while the improvements on the simpler MNIST and FashionMNIST problems are less pronounced. For the simpler problems, the twish function likely implements a Swish-like solution. In contrast, for the more complex CIFAR-10 dataset, it appears to exploit its greater expressive power to outperform the other activation functions.

Table 2: Average number of epochs required to reach 90% of the total accuracy span achieved during the entire training process.

| AF | MNIST | FashionMNIST | CIFAR-10 |
|---|---|---|---|
| relu | $9.5 \pm 4.0$ | $4.5 \pm 1.2$ | $8.0 \pm 1.7$ |
| prelu | $9.7 \pm 5.0$ | $5.1 \pm 1.3$ | $7.5 \pm 1.7$ |
| elu | $24.9 \pm 8.7$ | $5.6 \pm 1.3$ | $25.3 \pm 7.1$ |
| selu | - | $6.5 \pm 1.8$ | - |
| gelu | $7.8 \pm 3.8$ | $4.5 \pm 1.1$ | $7.1 \pm 1.3$ |
| mish | $9.4 \pm 4.4$ | $5.0 \pm 1.3$ | $8.7 \pm 1.8$ |
| swish | $7.0 \pm 3.0$ | $\mathbf{4.4 \pm 1.0}$ | $7.2 \pm 1.5$ |
| twish | $\mathbf{5.8 \pm 1.7}$ | $4.8 \pm 1.4$ | $\mathbf{6.4 \pm 1.1}$ |

Next, Table 2 analyzes the convergence time. We measure the average number of epochs required to reach 90% of the total accuracy span achieved during the entire training process[1]. Consistent with our previous results, the twish function generally shows faster convergence times. The only exception is FashionMNIST, where Swish reaches the convergence point slightly earlier. However, the differences among all four activations are not statistically significant for this dataset, with twish closely following Swish. Networks trained with the SELU activation function failed to reach the convergence point after 30 training epochs for the MNIST and CIFAR-10 problems.

Overall, our experiments demonstrate that the twish activation function consistently outperforms other popular activations like ReLU, pReLU, and Swish across several datasets. We have shown that twish not only achieves higher final accuracy, especially on the more complex CIFAR-10 dataset, but also leads to faster convergence during training. These results suggest that the enhanced expressive power of the twish function, derived from its learned parametric form, allows it to adapt more effectively to the complexities of different problems, offering a clear advantage over conventional activation functions.

## 5 CONCLUSIONS

This study has introduced a novel, data-driven methodology for discovering new activation functions, challenging the traditional reliance on manually designed functions or simple parametric searches. By modeling the activation function as a small neural network—a Self-Learning Activation Function (SLAF)—we were able to train a diverse collection of functional shapes. Our core contribution lies in this unique discovery process: using Principal Component Analysis (PCA) on these learned functions to identify their main functional modes. This approach allowed us to derive a new, powerful activation function, which we termed twish, by fitting simple analytical forms to the principal components. This methodology offers a more streamlined and less biased alternative to extensive parametric searches, simplifying the process of finding new and effective non-linearities for neural networks.

The experimental results presented here provide strong evidence for the effectiveness of the twish function. Our comparative analysis showed that twish consistently outperformed other widely-used activation functions, including ReLU, pReLU, and Swish, in terms of both final accuracy and convergence speed. This was particularly evident on the more complex CIFAR-10 dataset, where twish demonstrated a significant advantage. The superior performance of twish suggests that the expressive power of its learned form allows it to adapt more effectively to the complexities of a given problem, offering a clear advantage over conventional, fixed-form functions. The fact that twish generalizes the Swish function—which was itself the result of a complex automated search—highlights the strength and potential of our discovery method.

---

[1]The convergence point is defined as the epoch where the test accuracy first exceeds $acc_{min} + 0.9(acc_{max} - acc_{min})$, where $acc_{min}$ and $acc_{max}$ are the minimum and maximum accuracy values observed during training.

While our findings are promising, it is important to acknowledge the limitations of this initial study. Our experiments were conducted on a restricted set of relatively small-scale datasets (MNIST, FashionMNIST, and CIFAR-10) and using simple feedforward and convolutional neural network architectures. Consequently, the results presented here are not directly comparable to the state-of-the-art performance on these benchmarks, which often rely on much deeper and more complex models. The primary goal of this work was not to set new performance records, but rather to validate our novel method for activation function discovery and to introduce the twish function.

Despite these limitations, our work serves as a proof of concept for a new and promising approach to functional discovery in deep learning. We believe this methodology can be a valuable tool for future research. We encourage the broader scientific community to experiment with the twish function on a wider range of architectures and datasets. Further experimentation will be crucial to fully validate its effectiveness and to explore its potential in more advanced deep learning applications.

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

## A    AUTHORS STATEMENT ON THE USE OF LARGE LANGUAGE MODELS (LLMs)

This work was conducted with the assistance of a large language model (LLM), specifically Gemini, used as a general-purpose writing and ideation tool. The role of the LLM was significant enough to warrant disclosure.

The LLM was used primarily to enhance the clarity, structure, and flow of the manuscript. Its contributions include:

- Refining and organizing the text: The model was used to improve sentence structure, grammar, and overall coherence across various sections, including the introduction and conclusions. It helped restructure disorganized ideas into a logical narrative that is easier for readers to follow.

- Improving academic tone: The LLM was prompted to formalize the language and terminology to align with the standards of scientific writing, ensuring that the manuscript's tone is professional and precise.

- Reviewing and editing: The model was used to review text sections, identify redundancies, and propose alternative phrasing for conciseness and impact. It also helped to rephrase specific sentences to more accurately reflect the authors' intended meaning.

It is important to note that the LLM did not perform any original research, data analysis, or interpret experimental results. All core ideas, experimental designs, and data interpretations are exclusively the intellectual property of the authors. The LLM acted just as a sophisticated writing and editing assistant, similar to how a human co-author might provide feedback on prose and structure.

