# OpenReview forum: "Learning activation functions with PCA on a set of diverse piecewise-linear self-trained mappings"
_ICLR.cc/2026/Conference — Submitted to ICLR 2026_

### Official Review · Reviewer_WARg · 2025-10-26

**Soundness:** 2
**Presentation:** 3
**Contribution:** 2
**Rating:** 2
**Confidence:** 4

**Summary:**

This submission proposes a method to find new formulations for activation functions. Motivated by previous work, that proposes parameterized activation functions, they propose a two step approach. In the first step, they express activation functions as MLPs - based on the universal function approximation theorem. In a second step, they identify the eigenfunction within the learned activation functions, and find a compressed symbolic form with few learned parameters. Empirically, they demonstrate that their activation function performs well on small computer vision datasets.

**Strengths:**

- The topic of the submission is very interesting: many aspects of deep learning architectures are iteratively designed to address specific shortcomings. The idea of learning activation functions and then condensing them into efficient and powerful versions is appealing and fits well into the general learning regime of DL.
- The idea of representing a large class of activation functions with an MLP is neat. I’d be very curious to compare the performance and behavior of models with such learned activation functions to regular models and encourage the authors to add such an analysis.
- The identification of eigenfunctions seems a powerful idea in this application. Given that the learned activations are piecewise-linear, the identification appears tractable. While this is a nice idea, the lack of details in this part makes it challenging to evaluate.

**Weaknesses:**

While I appreciate the general idea of the submission, there are several weaknesses.
- **W1 - experimental evaluation:** The experimental identification of activation functions is based on small and simple image classification problems with 2-layer MLPs. While the evaluation adds a small CNN, both the basis to activation function search as well as the evaluation are very limited. I understand that the method requires repeated training of SLAF models to identify eigenfunction with a corresponding computational burden. However, if the authors are convinced of the merit of their method, they need to evaluate on larger and different domains, different architectures and different tasks in order to claim any general benefit.
- **W2 - eigenfunction identification:** The method of eigenfunction identification remains unclear. Since that’s at the core of the proposed method and there is ample space within the page limit, the gap is problematic. I strongly encourage the authors to include further details on how they identify eigenfunctions and what function space they consider.
- **W3 - inductive biases of SLAF:** The expression of the learned activation functions as MLPs is elegant and based in the universal function approximation theorem. That said, related work has shown that NNs generally and MLPs specifically have a bias towards specific parts of the signal, e.g., https://arxiv.org/abs/2403.02241. By induction, the MLP that the authors used for their search inherits that bias, and so it’s not entirely surprising that the activation function that is found is not dissimilar to existing functions. I encourage the authors to discuss these biases, whether or not they are desirable, and how that affects the overall search space.

**Questions:**

How are the eigenfunctions identified? Are you searching over a specific function space? If not, how did you to identify this particular formulation?

---

> ### Author Response · Authors · 2025-12-01
>
> We thank the reviewer for the comments and feedback, and particularly appreciate his positive assessment of the strengths of our contribution.
>
> Regarding the identified weaknesses, we would first like to mention that we are fully conscious of the limited scope of our experimental evaluation, as we previously acknowledged in the manuscript. In fact, we are actively working on new experiments to test the generalizability of the novel twish activation function using more complex deep learning models, such as ResNet trained on ImageNet and LLMs. However, the results will not be available in time for inclusion in the revised manuscript before the deadline.
>
> The eigenfunctions were obtained by directly applying PCA to the trained activation functions. We first evaluated the trained SLAFs at 1,600 equally spaced points within the range $[-4, 4]$. This range was selected to cover most of the input values that the neurons effectively encounter during training (as detailed in the response to Reviewer QDUi). PCA was subsequently applied to these 4,608 vectors (one for each trained activation function), each of which was 1,600-dimensional. Consequently, each principal component is also a 1,600-dimensional vector. We found by visual inspection that the proposed functional form, $x \tanh (\beta x) + \gamma x$, provides a good fit to the first two principal components.
>
> Finally, we agree that there might be some biases associated with the use of the ReLU function as the basis activation within the SLAFs. If a different activation, e.g., $\tanh$, is chosen, we observe more complex functional shapes for the resulting principal components and, generally, a degraded generalization capability. This observation could, of course, be linked to the inductive bias towards simpler and smoother functions that is reported for the ReLU activation in the suggested paper.

---

### Official Review · Reviewer_pzcB · 2025-10-30

**Soundness:** 3
**Presentation:** 3
**Contribution:** 3
**Rating:** 8
**Confidence:** 3

**Summary:**

This paper proposes a novel, data-driven methodology for discovering new activation functions. Instead of manual design or simple parametric search, the authors first define a "Self-Learning Activation Function" (SLAF) as a small, one-hidden-layer MLP, which is effectively a flexible piecewise-linear function. They then train thousands of networks (4608 in total) embedded with these SLAFs on a range of problems (MNIST, Fashion-MNIST, CIFAR-10) and SLAF network sizes to generate a diverse collection of effective activation functions.

The core contribution is the analysis of this collection. The authors apply Principal Component Analysis (PCA) and find that the first two principal components (PCs) explain over 99.5% of the variance in the learned functions. These two PCs are well-approximated by simple analytical functions. The first component is described as 'x * tanh(beta * x)' (a soft absolute value). The second component is a simple linear function, described as 'gamma * x'.

By combining these two components, the authors derive a new, two-parameter learnable activation function, which they term twish. It is defined by the expression 'x * tanh(beta * x) + gamma * x'. The authors note that twish is a generalization of the Swish activation function. In validation experiments on simple CNNs, twish is shown to consistently outperform ReLU, pReLU, and Swish in terms of both final test accuracy (particularly on the more complex CIFAR-10 dataset) and convergence speed. The work serves as a strong proof-of-concept for this new PCA-based discovery method.

**Strengths:**

Novelty of the Discovery Method: The core strength is the PCA-based methodology. Using PCA on a large ensemble of learned functions (the SLAFs) to distill their essential components is a highly original and intelligent approach to functional discovery.

Strong Empirical Finding: The discovery that just two principal components explain >99.5% of the variance is a powerful and elegant result, strongly suggesting a simple, low-dimensional underlying structure for effective activation functions.

**Weaknesses:**

Limited Scale of Validation: This is the primary weakness, which the authors rightly acknowledge. The validation experiments (Section 4) use small datasets (MNIST, CIFAR-10) and very simple CNNs. The true test of a new activation function is its performance and stability in deep, complex models (e.g., ResNets, ViTs) on large-scale tasks (e.g., ImageNet, large NLP corpora). Without this, it's hard to judge if "twish" will be broadly useful.

Limited Diversity of SLAF Training: The initial 4608 SLAFs were all trained on simple FFNs for small-scale image classification. It's an open question whether this set is "diverse" enough. The discovered PCs might be biased towards this specific task and architecture family.

**Questions:**

1. How were the beta and gamma parameters of the twish function initialized in the Section 4 experiments? Did you observe any sensitivity to this initialization during training?

2. Your analysis was based on SLAFs trained for classification tasks. Did you also conduct similar experiments for regression tasks?
If not, would you expect the PCA analysis on SLAFs trained for regression (e.g., optimizing MSE) to reveal the same principal components and, consequently, the same "twish" function, or do you think different functional forms might emerge?

---

> ### Author Response · Authors · 2025-12-01
>
> We thank the reviewer for the positive assessment. We are aware of the mentioned weaknesses and are actively working on new experiments to test the generality of the new twish activation function. We are currently performing tests with more complex deep learning models, such as ResNet trained on ImageNet and LLMs.
>
> In response to the questions raised, the parameters $\beta$ and $\gamma$ are initialized in all cases so that the initial shape of the activation function is a swish.
>
> We have not carried out experiments on regression problems, but we will consider including them in future research. Nevertheless, we expect that the functional form is more related to the basis activation function used within the SLAFs (a ReLU in our experiments) than to the type of problem being solved.

---

### Official Review · Reviewer_z1kh · 2025-10-31

**Soundness:** 2
**Presentation:** 2
**Contribution:** 1
**Rating:** 2
**Confidence:** 4

**Summary:**

The paper proposes a data-driven method to discover new neural network activation functions. It defines a small neural subnetwork (SLAF) that learns its own activation mapping during training, and then applies Principal Component Analysis (PCA) to thousands of these learned functions trained on small, classical datasets (MNIST, FashionMNIST, and CIFAR-10). The claim is that the first two principal components explain nearly all the variation and lead to a new function, they call twish, defined as $f(x; \beta, \gamma) = x \tanh(\beta x) + \gamma x$, which is a slight generalization of the Swish activation. Experiments are conducted on simple convolutional neural networks and results illustrating marginally better accuracy and faster convergence compared with ReLU, PReLU, and Swish are presented.

**Strengths:**

The paper is fairly clearly written.

**Weaknesses:**

This work presents a slight generalization of a popular activation function as the key contribution. Despite the work being experimental in nature, the conclusions as to the value of this new activation function are drawn based on toy datasets (MNIST, FashionMNIST and CIFAR10). In my view the experiments are not adequate to make any substantive claims and for this paper to be interesting I think you would need very strong evidence. I also don't see why the ideas behind the derivation of this activation function are particularly new, there are many techniques and approaches for deriving activation functions. In addition, despite the supposed benefit being that we derive activations from data, the key finding appears to be something close to what we already use.

**Questions:**

- In what sense is applying PCA to a collection of learned functions conceptually new, compared to previous meta-learning or search-based activation discovery frameworks, e.g., the one from Ramachandran et al?

- What theoretical advantage does twish offer over existing parameterized activations (e.g., Swish, Mish, GELU), for instance from an edge of chaos perspective / dynamical isometry / vanishing and exploding gradients perspective?

---

> ### Author Response · Authors · 2025-12-01
>
> We thank the anonymous reviewer for the comments and feedback. We are conscious of the limited scope of our experiments, as we stated in the manuscript. We are currently performing tests with more complex deep learning models, such as ResNet trained on ImageNet and LLMs, to test the generalizability of our approach. However, the results will not be available in time for inclusion in the revised manuscript before the deadline.
>
> The main novelty of our contribution lies in the approach to activation function discovery, which avoids possible biases inherent to specific functions used in a purely parametric search. Furthermore, the fact that the resulting activation is quite close to functions obtained via different search methods should be viewed as a strength of the methodology, rather than a weakness. This result might also indicate a possible optimality of swish-like functions.

---

### Official Review · Reviewer_QDUi · 2025-11-01

**Soundness:** 3
**Presentation:** 3
**Contribution:** 2
**Rating:** 2
**Confidence:** 3

**Summary:**

This paper proposes a data-driven method for discovering activation functions.
A simple ReLU-based MLP is trained as a Self-Learned Activation Function (SLAF), and the shapes of the resulting functions — obtained from various datasets and network widths — are analyzed using PCA.
The top two principal components explain ~99.5% of the variance, corresponding respectively to $x\tanh(\beta x)$ and $\gamma x$.
Building on this observation, the authors define a new activation function Twish,
$f(x; \beta, \gamma) = x\tanh(\beta x) + \gamma x$.
Experimental results show that Twish achieves faster convergence and higher accuracy than ReLU, pReLU, and Swish across benchmark datasets such as MNIST, FashionMNIST, and CIFAR-10.

**Strengths:**

- Data-driven discovery: Uses PCA over a large pool of learned SLAFs to quantify shape diversity and reveal a compact, interpretable 2-D structure of activation functions.
-Clear, reproducible pipeline: SLAF training → uniform sampling on a fixed grid → PCA → analytic fitting of PCs → definition of Twish. simple and easy to replicate.
- Strong quantitative support: The first two principal components explain ~99.5% of the variance, providing a solid justification for dimensionality reduction.
- Empirical signal: On small CNNs (e.g., CIFAR-10), Twish shows faster convergence and consistently higher accuracy that ReLU, pReLU, and Swish.
- Practical value: Offers a lightweight alternative to large parameter searches for activation design, turning observed functional modes into a compact parametric family.

**Weaknesses:**

- Constrained search space: SLAFs are ReLU-based and thus piecewise-linear; the approach may bias discoveries toward piecewise linear shapes and under-explore smooth families.
- Preprocessing dependence: PCA is performed on BN-normalized pre-activations sampled only on [−4,4];  stability to the choice of range/resolution/normalization is not established.
- Limited scale: Experiments focus on small datasets and shallow models; generalization to ResNet/ViT/ImageNet or transformer LMs remains untested.
- Baseline coverage: Direct, controlled comparisons against modern smooth activations (e.g., GLEU, Mish, ELU, SELU) under the same setup are missing.
- Implementation specifics: The sharing/initialization/constraints of the learnable parameters $(\beta, \gamma)$ (layer, channel, or unit-level) are insufficiently detailed for full reproducibility.

**Questions:**

1. Where are $(\beta, \gamma)$ shared - at the layer, channel, or neuron level? Do you impose any constraints (e.g., $\beta \geq 0$)?
2. How do the PCA results and Twish’s performance change if BatchNorm is removed or if a different input range is used for sampling?
3. If the SLAF is trained as a non–piecewise-linear network (e.g., using tanh or RBF bases), does the same principal-component structure persist?
4. Do you have results applying Twish to larger models (e.g., ResNet, ViT, Transformers) or large-scale datasets (e.g., ImageNet)?
5. You exclude ELU/SELU/GELU citing Ramachandran et al. (2017) showing Swish's superiority. However, to rule out setting dependence, shouldn't you still report direct, controlled comparisons under your exact setup or at least provide their PCA coordinates relative to Twish?

---

> ### Author Response · Authors · 2025-12-01
>
> We thank the anonymous reviewer for the thorough comments and feedback, which will certainly help us improve our work. In the following, we address the raised concerns and provide a response to all the questions.
>
> First, we would like to clarify that, as we already acknowledge in the manuscript, we are aware of the limited scope of our experiments. We are currently performing tests with more complex deep learning models, such as ResNet trained on ImageNet and LLMs. However, the results will not be available in time for inclusion in the revised manuscript before the deadline.
>
> The twish parameters ($\beta$ and $\gamma$) are shared across all neurons in the network, and they are initialized such that the initial shape of the activation function matches a swish. No explicit constraints are imposed on their values.
>
> Batch Normalization is applied with non-trainable parameters. Specifically, the batch pre-activations are normalized to a $\mathcal{N}(0, 1)$ distribution. Given this input scale, we believe that sampling in the range $[-4, 4]$ is sufficient to cover most of the input values that the neurons effectively encounter during training.
>
> The observed solution will certainly change if we use bounded activation functions (such as $\tanh$ or $\text{RBF}$) as the basis for the SLAFs. The choice of ReLU is due to its inherent simplicity and the inductive bias towards smoother functions that it introduces, a point also raised by Reviewer WARg.
>
> Although further experimentation is required, some preliminary tests using the $\tanh$ activation function have shown more complex functional shapes and a degraded generalization capability.

---

> > ### Author Response · Authors · 2025-12-03
> >
> > Following the reviewer's suggestion, we have also performed new tests with other activation functions, namely ELU, SELU, GELU, and Mish. The results have been included in Tables 1 and 2 of the revised version and are consistent with our previous observations. The new activation functions underperformed compared to our original approach, both in terms of test accuracy at the end of the 30 training epochs (Table 1) and the average number of epochs required to reach 90% of the accuracy span (Table 2).

---

### Meta-Review · Area_Chair_ysGx · 2026-01-04

**Summary:**

The submission presents a method for learning activation functions that involves training a large number of small MLPs as initial activation functions, then uses PCA on discretized versions of these MLPs as the basis for discovering symbolic representations of these activation functions. The reviewers generally appreciated the clarity of writing and the inventiveness of the approach. However, they were consistent in pointing out that the approach is empirically motivated but relies only on small-scale datasets and network architectures. The diversity of the SLAF models was also questioned: these currently all use ReLU activation functions, which may have an impact on the meta-learning process. It was also pointed out that the discovered (symbolic) activation functions are already quite similar to existing activation functions, in terms of both their symbolic form and the performance of the resulting models.

**Reviewer Concerns:**

I believe that most of the concerns related to experimental setup clarifications and algorithm details have been sufficiently addressed in the rebuttal. However, the main concerns I have with this submission are: (i) the scope of the experimental validation (datasets and network size); and (ii) the lack of investigation into the sensitivity of the SLAFs activation functions. I would encourage the authors to investigate these two directions in a future submission by, e.g.: (i) demonstrating that the meta-learning process can be conducted on small-scale datasets/networks and that the resulting activation functions generalise to large setups; and (ii) showing a different choice of activation function in the SLAFs does/doesn't lead to dramatically different discovered functions. You may also consider iterating the meta-learning process by using the discovered activation function inside the SLAFs.

**Reviewer Scores:**

I do not think the reviewers would have changed their scores substantially, as the scope of the rebuttals was mostly directed at the more minor concerns raised by each reviewer.

---

### Decision · Program_Chairs · 2026-01-26

Reject